# Nutrition Status and Renal Function as Predictors in Acute Myocardial Infarction with and without Cancer: A Single Center Retrospective Study

**DOI:** 10.3390/nu13082663

**Published:** 2021-07-30

**Authors:** Naoki Itaya, Ako Fukami, Tatsuyuki Kakuma, Yoshihiro Fukumoto

**Affiliations:** 1Division of Cardiovascular Medicine, Department of Internal Medicine, Kurume University School of Medicine, Kurume 830-0011, Japan; itaya_naoki@kurume-u.ac.jp (N.I.); fukami_ako@kurume-u.ac.jp (A.F.); 2Biostatistics Center, Kurume University, Kurume 830-0011, Japan; tkakuma@med.kurume-u.ac.jp

**Keywords:** onco-cardiology, nutrition status, cancer, acute myocardial infarction

## Abstract

***Background:*** Clinical characteristics of nutrition status in acute myocardial infarction (AMI) patients with cancer remains unknown. Therefore, this study aimed to clarify the differences of clinical parameters, including nutrition status, between AMI patients with and without history of cancer. ***Methods and Results:*** This retrospective cohort study, using the database of AMI between 2014 and 2019 in Kurume University Hospital, enrolled 411 patients; AMI patients without cancer (*n* = 358, 87.1%) and with cancer (*n* = 53, 12.9%). AMI patients with cancer were significantly older with lower body weight, worse renal function, and worse nutrition status. Next, we divided the patients into 4 groups by cancer, age, and plaque area, detected by coronary image devices. The prediction model indicated that nutrition, lipid, and renal functions were significant predictors of AMI with cancer. The ordinal logistic regression model revealed that worse nutrition status, renal dysfunction, lower uric acid, and elevated blood pressure were significant predictors. Finally, we were able to calculate the probability of the presence of cancer, by combining each factor and scoring. ***Conclusions:*** Worse nutrition status and renal dysfunction were associated with AMI with cancer, in which nutrition status was a major different characteristic from those without cancer.

## 1. Introduction

Acute myocardial infarction (AMI) is mainly caused by coronary arteries thrombus due to atherosclerotic plaque rupture or endothelial erosion, and sometimes coronary artery spasm, microvascular thrombus, or others [1]. The formation of fibrofatty lesions in the atherosclerotic vulnerable plaques occurs during the atherosclerosis progression [2], which have been treated by lipid-lowering therapy to stabilize during these 2 decades [3,4]. Further, it has been reported that rapid plaque progression of moderately severe vulnerable plaques is the critical step prior to AMI in most patients [5,6,7]. Actually, accumulating evidence demonstrates that atherosclerotic risk factors, including hypertension, dyslipidemia, diabetes mellitus, and cigarette smoking, develop atherosclerotic lesions through immune system [2]. The atherosclerotic plaques often contain lipid core and fibrous cap, which consist of smooth muscle cells, macrophages, angiogenesis, and adventitial inflammation [8]. Taken together, when focused on vulnerable coronary atherosclerosis, not stable atheroma, it is considered that AMI most likely occurs due to rapidly progressed coronary atherosclerosis, caused by traditional atherosclerotic risk factors, including age; however, even small atherosclerotic plaques can cause AMI in some patients, which might be associated with some other risk factors.

In cancer patients, the mechanisms of atherosclerosis progression might be different. Especially, clonal hematopoiesis with indeterminate potential (CHIP) has been reported to increase the risk of AMI [9]. CHIP carriers had a 4.0-fold (95% confidence interval 2.4–6.7) higher risk of myocardial infarction than non-carriers [9]. It has been also reported that each of DNA methyltransferase 3A (DNMT3A), ten-eleven translocation-2 (TET2), additional sex combs-like 1 (ASXL1), and Janus kinase 2 (JAK2) mutation was associated with coronary artery disease, and that CHIP carriers with these mutations also had high levels of coronary artery calcification [9]. Further, TET2-deficient bone marrow cells enlarge atherosclerotic plaques by infiltrating macrophages, and TET2-deficiency increases cytokines/chemokines such as C-X-C motif ligand (CXCL)-1, CXCL 2, CXCL 3, interleukin (IL)-6 and IL-1β [10]. Also, Heyde et al., have reported that CHIP causes hematopoietic stem cell proliferation, promotes arteriosclerosis, and leads to a vicious cycle of proliferation of hematopoietic stem cells [11]. Above all, CHIP is an independent coronary risk factor in patients with premature menopause, especially in patients with spontaneous premature menopause [12]. Thus, coronary risk factors in the clinical settings might be different between cancer and non-cancer patients.

During these 4 decades, cancer is the first cause of death in Japan, and cardiovascular diseases (CVDs) are the second [13]. Because of recent improvement of cancer prognosis due to the advancement of cancer early detection, surgery, and anticancer drug treatment, the number of cancer survivors has increased [6,7]. Cancer and CVDs have common risks of lifestyle factors, such as smoking, obesity, and unhealthy food intake, in which healthy lifestyle is oppositely associated with a longer life expectancy free from major chronic diseases, including cancer and CVDs [14]. However, the clinical characteristics, especially in the aspects of nutrition status, remains scant in AMI patients with cancer.

Therefore, the purpose of the present study was to clarify the differences of clinical parameters in AMI with and without history of cancer, focusing on traditional coronary risk factors and nutrition status, and to develop a statistical model to evaluate the presence of cancer in patients with AMI, using database of AMI in Kurume University Hospital.

## 2. Materials and Methods

### 2.1. Study Design

This study was retrospective cohort study using the database of AMI in Cardiovascular Medicine, Kurume University Hospital. We enrolled 437 patients, who were treated by primary percutaneous coronary intervention (PCI) due to AMI in Cardiovascular Medicine, Kurume University Hospital from January 2014 to December 2019. We excluded patients if the patients had following reasons; (1) AMI occurred due to PCI complication, (2) PCI due to acute stent thrombosis, (3) unsuccessful PCI, and (4) coronary bypass surgery due to multi-vessel coronary artery disease.

### 2.2. Data Collection

Baseline demographic data were collected based on the medical records, including age, sex, height, body weight, waist, medications, traditional risk factors (hypertension, glucose intolerance/diabetes mellitus and dyslipidemia), blood pressure (BP), pulse rate, heart rate, and comorbidities (coronary artery disease, hypertensive heart disease, cardiomyopathy, valvular heart diseases, and congenital heart diseases). All cardiovascular diseases were diagnosed by expert cardiologists. AMI was diagnosed according to fourth universal definition of myocardial infarction [15].

All patients were treated by PCI combined with the use of image devices, such as intravascular ultrasound (IVUS) or optical coherence tomography (OCT). We retrospectively measured the coronary atherosclerotic plaque area by image devices at the culprit lesions.

### 2.3. Blood Sampling

Peripheral blood was drawn from the antecubital vein for measurements of blood cell counts, lipid profiles including total cholesterol (T.chol), low-density lipoprotein cholesterol (LDL-c), high-density lipoprotein cholesterol (HDL-c), and triglyceride, liver and renal function markers including creatinine (Cr) and estimated glomerular filtration rate (eGFR), glycemic parameters of fasting plasma glucose (FPG) and hemoglobin A1c (HbA1c), uric acid, troponin, and N-terminal pro-brain natriuretic peptide (NT-pro-BNP). These chemistries were measured at a commercially available laboratory in Kurume University Hospital.

### 2.4. Definition of Comorbidities

Hypertension was defined as the use of antihypertensive drugs and/or systolic blood pressure ≥ 140 mmHg or diastolic blood pressure ≥ 90 mmHg. Dyslipidemia was defined as the use of lipid-lowering drugs and/or plasma LDL-c ≥ 140 mg/dL and/or triglycerides ≥ 150 mg/dL, and/or HDL-c < 40 mg/dL. Diabetes mellitus was diagnosed using antidiabetic drugs and/or fasting plasma glucose ≥ 110 mg/dL or HbA_1c_ ≥ 6.5%.

### 2.5. Evaluation of Nutrition Status

The nutrition status was evaluated using the Glasgow Prognostic Score (GPS) [16], modified GPS (mGPS) [17,18] and Controlling Nutritional Status (CONUT) [19]. 

### 2.6. Statistical Analysis

All continuous variables were given as the mean ± standard deviations (SDs) or median with interquartile range. Categorical data were presented as number (*n*) or percentage (%). For intergroup univariate comparisons, an unpaired *t* test was applied in continuous variables and chi-square test in categorical variables.

The mean ± SDs and frequencies were presented by the two groups with and without cancer (Table 1 and Table 2).

To evaluate the impact of various risk factors of myocardial infarction on cancer and non-cancer patients, following three analytical steps were taken. First, the classification and regression tree (CART) model was employed to define risk groups (Table 3). Cancer status (yes or no) was used as the response variable and plaque area, age and sex were used as predictors. The fitted probability for the levels of the response was calculated, and the split is chosen to minimize the residual log-likelihood chi-square. Second, principal component analyses were performed to derive synthetic variables based on five sets of risk factors. Specifically, three measurements of nutrition, two measurements for lipid, glucose, blood pressure and renal function were subjected in the principal component analyses which render a way to avoid collinearity problems among highly correlated measurements within each set of risk factor (Table 4 and Table 5). Finally, ordinal logistic regression was employed to evaluate effect of each synthetic variables on the risk groups derived from the CART. Effect of each synthetic variable was interpreted based on the odds ratio (Table 6 and Table 7), and the predicted probability of the risk groups were calculated (Figure 5). To compare the results with logistic regressing model with cancer/non-cancer as response variable, an additional table was added (Table 8).

Statistical significance was defined as *p* value < 0.05. All statistical analyses were performed using JMP Pro 13.0 and SAS software (Release 9.3; SAS Institute, Cary, NC, USA).

### 2.7. Description of Ordinal Logistic Regression Model 

Each subject denoted by *i* (*i* = 1, …, *N)* is classified into risk group denoted as “G1”, “G2”, “G3” or “G4”, and group membership is represented by random variable *Y*. *Y* takes value = 1 if G1, *Y* = 2 if G2, *Y* = 3 if G3 and *Y* = 3 if G4. Ordinal logistic regression is used to model$\;Y_{i}$. Let *X* be a vector of covariate defined as $X={\left(X_{1},\cdots ,X_{6},W\right)}^{\prime }$ where $X_{1},\cdots ,X_{6}$ are 6 risk variables. W is a dummy variable, which takes value 0 or 1 for non-smoker and smoker respectively. Given covariate vector *X*, cumulative probability of *Y* is written as P(Y≤k|X), and ordinal logistic regression model is defined as
(1)$$\mathrm{logit}\left(P\left(Y\leq k|X\right)\right)=log\left\{\frac{P\left(Y\leq k|X\right)}{1-P\left(Y\leq k|X\right)}\right\}=\alpha_{k}-X^{\prime }\beta$$
where $\alpha_{k}$ is an intercept with $\alpha_{4}=0$, and β is 7×1 parameter vector. From definition of the model, the expected predicted probabilities for G1, G2, G3 and G4 are given by
(2)$$\;\;\;\;\;\;\;\;\;\;\;\;\;\;\;\;\;\;\;\;\;\;\;\;P_{1}=P\left(Y=1|X\right)=\frac{exp\left(\alpha_{1}-X^{\prime }\beta \right)}{1-exp\left(\alpha_{1}-X^{\prime }\beta \right)}\;,\;\;\;\;\;\;\;\;\;\;\;\;\;\;\;\;\;\;\;\;\;\;\;\;\;P_{2}=P\left(Y=2|X\right)=\frac{exp\left(\alpha_{2}-X^{\prime }\beta \right)}{1-exp\left(\alpha_{2}-X^{\prime }\beta \right)}-\frac{exp\left(\alpha_{1}-X^{\prime }\beta \right)}{1-exp\left(\alpha_{1}-X^{\prime }\beta \right)}\;,\;\;\;\;\;\;\;\;\;\;\;\;\;\;\;\;\;\;\;\;\;\;\;\;\;\;P_{3}=P\left(Y=3|X\right)=\frac{exp\left(\alpha_{3}-X^{\prime }\beta \right)}{1-exp\left(\alpha_{3}-X^{\prime }\beta \right)}-\frac{exp\left(\alpha_{2}-X^{\prime }\beta \right)}{1-exp\left(\alpha_{2}-X^{\prime }\beta \right)},\;P_{4}=P\left(Y=4|X\right)=1-\frac{exp\left(\alpha_{3}-X^{\prime }\beta \right)}{1-exp\left(\alpha_{3}-X^{\prime }\beta \right)}$$

Finally, the predicted probabilities $\left({\hat{P}}_{1},{\hat{P}}_{2},{\hat{P}}_{3},{\hat{P}}_{4}\right)$ were obtained by plugging in parameter estimates $\left({\hat{\alpha }}_{k},\hat{\beta }\right)$ into the expected predicted probabilities, and ${\hat{\alpha }}_{k}-X^{\prime }\hat{\beta }$ is referred to as the linear predicted score in the Figure 5. 

Since our model is defined as $\mathrm{logit}\left(P\left(Y\leq k|X\right)\right)=\alpha_{k}-X^{\prime }\beta$, log odds ratio of belonging lower risk group when Nutrition score increase one unit is given by
(3)$$\mathrm{logit}\left(P\left(Y\leq k|X_{1}=a+1\right)\right)-\mathrm{logit}\left(P\left(Y\leq k|X_{1}=a\right)\right)=-\beta_{1}$$
where $\beta_{1}$ is the parameter estimate of “Nutrition”, thus, odds ratio is given by $\exp\;\left(-\beta_{1}\right).$ However, interpretation of odds ratio for a risk variable is easier for a subject being classified into higher risk group. To this end, we reversed the order of group membership (i.e., Y * = 5 − Y) and model Y * instead of Y where G1 was set as a reference group. Table 6 shown parameter estimates for modeling Y * where OR for nutrition is calculated as exp (0.232) = 1.26.

## 3. Results

### 3.1. Clinical Differences of Traditional Coronary Risk Factors and Nutrition Status with and without Cancer

Among 437 enrolled patients in the present study, 26 patients were excluded (Figure 1).

In the remaining 411 patients, there were 358 AMI patients without cancer (87.1%) and 53 with cancer (12.9%). To compare with AMI patients without cancer, those with cancer were significantly older with significantly lower body weight, lower diastolic blood pressure, anemia, worse renal function, lower albumin and cholesterol levels, and worse nutrition status, evaluated by CONUT score, GPS, and mGPS (Table 1).

Next in comorbidities, those with cancer had significantly higher prevalence of diabetes mellitus, chronic kidney disease, history of coronary artery bypass surgery treatment, and peripheral artery disease, who also had more frequent drug interventions by clopidogrel, ticlopidine, other antiplatelet agents, anti-cancer agents, and steroid (Table 2).

### 3.2. Risk Model of Cancer

To divide the patients into some groups to evaluate the presence of cancer, we first examined the association between age and coronary atherosclerotic plaque area (Figure 2).

Obviously, there was no cancer patient in Age < 60 years-old group. Then, the CART divided the patients by plaque area < 9.39 mm^2^ and those ≥9.39 mm^2^, which was the cut-off value of the presence and absence of cancer. According to this grouping process, we divided the whole patients into 4 groups to develop a new response variable to evaluate the presence of cancer (Figure 3 and Table 3). As we considered that if AMI occurred in patients with smaller coronary atherosclerotic plaques, the patients had higher risks, we defined the 4 groups as described in Figure 3 and Table 3.

Next, we have developed the prediction model, which consisted of nutrition, lipid, glucose, blood pressure, and renal function (Table 4).

The equation of nutrition was expressed as “Nutrition = (−1.307) + 0.243 * CONUT + 0.898 * GPS + 0.846 * mGPS”. Similarly, other equations were expressed as “Lipid = (−4.947) + 0.018 * LDL-c + 0.015 * T.chol”, “Glucose = (−5.448) + 0.594 * HbA1c + 0.0092 * FPG”, “Blood pressure = (−5.559) + 0.038 * Pulse pressure + 0.027 * systolic BP”, “Renal function = (1.151) + 0.469 * Cr + (−0.025) * eGFR”. Using these equations, we performed analysis of variance, which indicated that nutrition, lipid, and renal functions were significant predictors of risk grouping (Table 5, Figure 4).

The odds of a subject being classified into higher risk group (i.e., group membership j (=1,2,3,4) increase) were 1.26 times when value of the synthetic variable “Nutrition” increased 1 point (Table 7).

Variables with odds ratio greater than 1 had similar interpretation. On the other hand, variables with odds ratio less than 1 had reverse effects. For example, when value of the synthetic variable “Lipid” increased 1 point, the odds of a subject being classified into higher risk group were 0.88 times compared with odds of being classified into lower risk group (Table 7). The additional logistic regression model, where cancer/non-cancer was used a response factor with age and plaque area as adjusting variables, showed the insignificant increase in nutrition (Table 8). Direction of effect of each synthetic variable could be also seen in the Table 5. Finally, we were able to calculate the probability of the presence of cancer, by combining each factor and scoring (Figure 5).

## 4. Discussion

The major findings of the present study were that (a) the prevalence of cancer in AMI patients was 13%, (b) AMI patients with cancer were older with worse nutrition status and renal dysfunction, (c) nutrition status and renal function were consistent predictors for obtaining cancer in AMI, and (d) we were able to calculate the probability of the presence of cancer, by combining each factor and scoring. To the best of our knowledge, this is the first study that provides the evidence of differential risk factors including nutrition status in AMI patients between with and without cancer in details.

### 4.1. Prevalence of Cancer in AMI

In the US National Inpatient Sample database between 2004 and 2014 report, among 6,563,255 AMI patients, there were 5,966,955 with no cancer, 186,604 with current cancer (2.8%), and 409,697 with a historical diagnosis of cancer (6.2%) [20]. Among 175,146 patients in Swedish registries of first AMI between 2001 and 2014, there were 16,237 patients (9.3%), who had received care for cancer in the 5 years before AMI [21]. The prevalence of cancer in AMI patients was 13% in the present study, which was relatively higher that these nationwide data, probably because the enrolled patients in the study might have complicated conditions such as being hospitalized in our University Hospital.

### 4.2. Differences of Traditional Risk Factors and Nutrition Status in AMI between with and without Cancer

Hypertension, diabetes, current smoking, family history, and dyslipidemia are well known as traditional risk factors in AMI [22], and it has been recently reported that aging, higher prevalence of dyslipidemia and hypertension, and lower prevalence of obesity, diabetes mellitus, and smoking were risk factors in breast cancer survivors compared with general female population [23]. However, the differential risk factors for AMI with cancer compared with non-cancer remains scant. The present study indicated for the first time that renal dysfunction and worse nutrition status were the strongest risk factors in AMI patients with cancer. Besides these factors, univariate analyses (Table 1) showed that AMI with cancer patients had significantly lower levels of albumin, T.chol, and LDL-c than those without cancer, which seemed to be caused by deterioration of nutrition status, which were different from traditional coronary risk factors.

### 4.3. Prediction and Scoring of Cancer in AMI

In the present study, we have developed formulas regarding nutrition status, lipid, glucose blood pressure, and renal function (Table 4). Using these formulas, we are able to avoid to evaluate several data separately, such as T.chol and LDL-c, or Cr and eGFR, and to consider these factors as whole. Further, we have established the predicted scoring system for AMI with cancer (Figure 5), which means that we are able to predict if the patients with AMI have cancer, by calculating this score. This might be useful in clinical settings to recognize the presence of cancer.

### 4.4. Limitations

Several limitations should be acknowledged in the present study. First, the present study was an observational retrospective cohort study from a single center. Thus, there might be some bias. Second, we were not able to divide cancer patients into active cancer and history of cancer, due to the limited number of patients with cancer. Also, we were not able to evaluate cancer stage in this study. Third, we were not able to distinguish coronary plaque rupture from coronary artery erosion as causes of AMI. Fourth, the enrolled patients in the present study were hospitalized in University Hospital, which may also cause some bias. Fifth, due to the limited number of enrolled patients, we were not able to examine sex-difference in the present study. Taken together, further investigations should be necessary in larger multi-center cohort studies.

## 5. Conclusions

The present study demonstrates that the prevalence of cancer in AMI was 13%, and that worse nutrition status and renal dysfunction were associated with AMI with cancer, in which nutrition status was a major different characteristic from non-cancer. Further, we have developed formulas to predict the presence of cancer in AMI.

## Figures and Tables

**Figure 1 nutrients-13-02663-f001:**
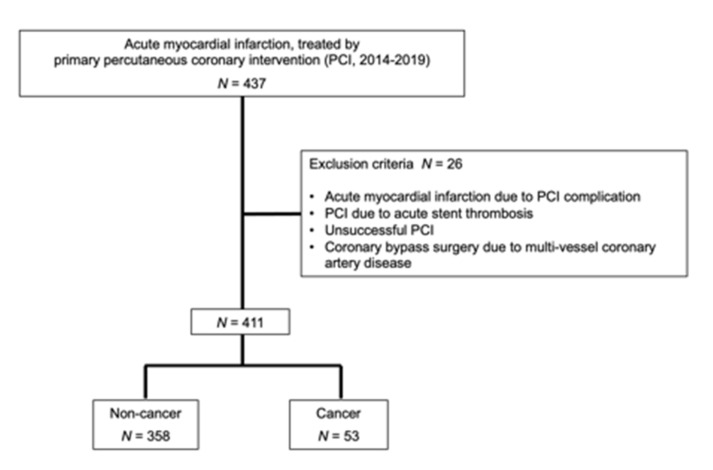
Enrollment of patients.

**Figure 2 nutrients-13-02663-f002:**
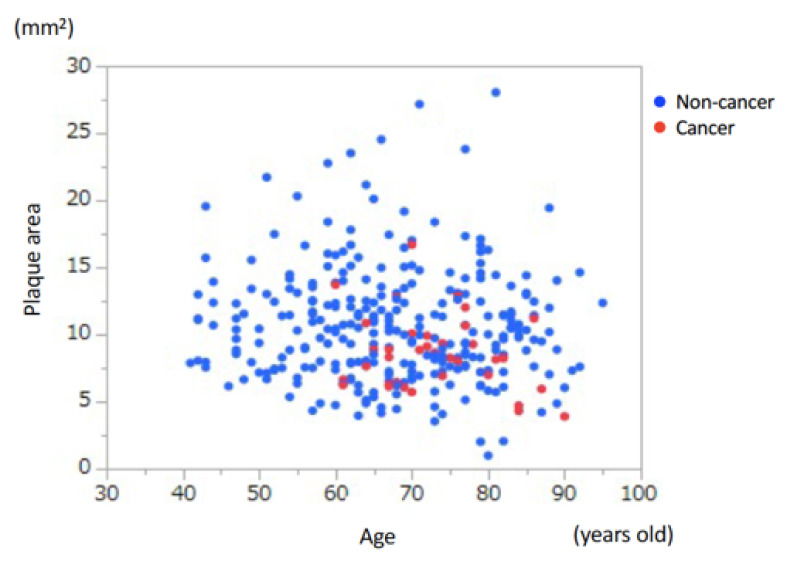
Association between age and coronary atherosclerotic plaque area. There was no significant association between age and coronary plaque area.

**Figure 3 nutrients-13-02663-f003:**
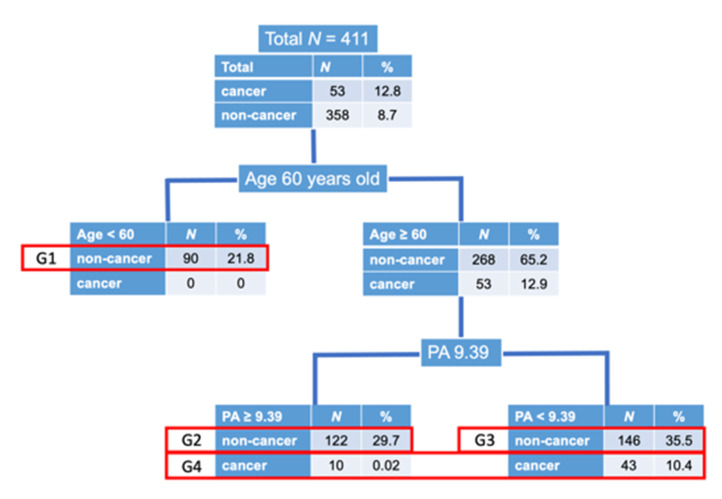
Disposition of patients. As there was no cancer patient in age < 60 years old, we first divided the patients into 2 groups according to age of 60 years old. Next, the CART defined that the cut-off value of coronary atherosclerotic plaque area (PA) to divide the patients into with and without cancer was 9.39 mm^2^. Then, we divided the patients ≥ 60 years old into 2 groups according to plaque area of 9.39 mm^2^, with and without cancer. Due to the small number of cancer patients, we made 4 groups as shown above. Group 1; non-cancer, age < 60 years old, Group 2; non-cancer, age ≥ 60 years old, plaque area ≥ 9.39 mm^2^, Group 3; non-cancer, age ≥ 60 years old, plaque area < 9.39 mm^2^, Group 4; cancer, age ≥ 60 years old.

**Figure 4 nutrients-13-02663-f004:**
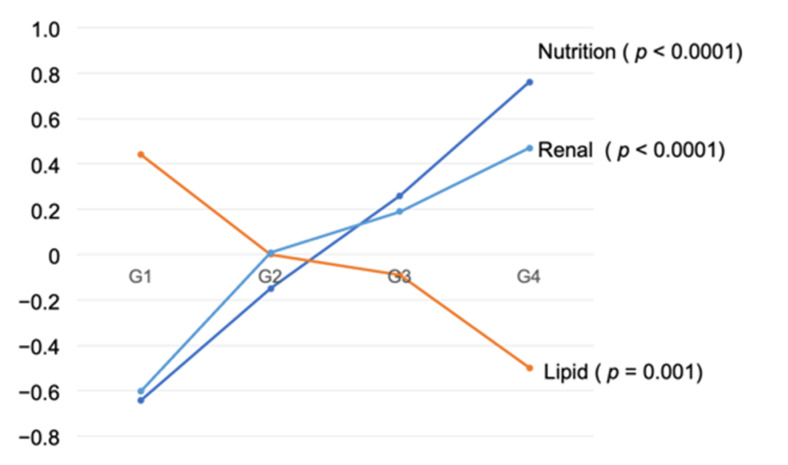
Mean of synthetic variable by Four Groups.

**Figure 5 nutrients-13-02663-f005:**
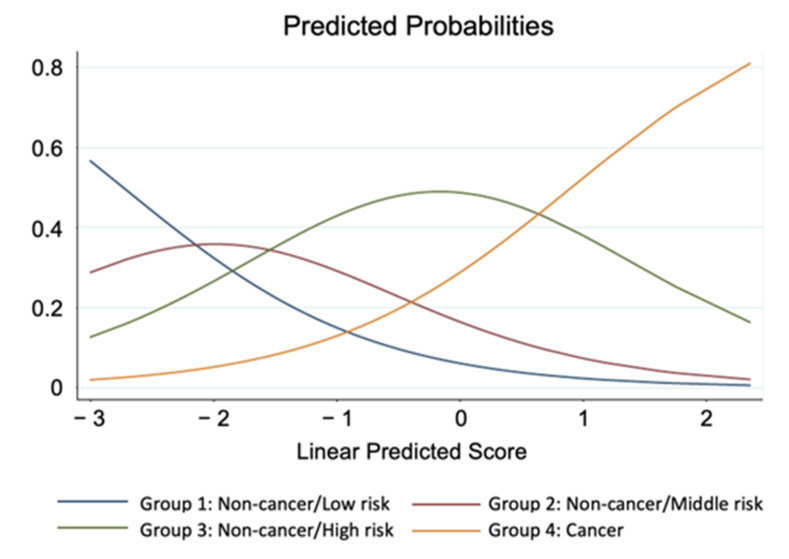
Predicted probabilities of classification to 4 risk groups. Linear predicted Score is referred to as linear combination of parameter estimates with its corresponding variable scores in the ordinal logistic regression model.

**Table 1 nutrients-13-02663-t001:** Clinical characteristics of acute myocardial infarction patients with and without cancer.

	Non-Cancer (*N* = 358)	Cancer (*N* = 53)	*p* Value
Age, y	67.6 ± 12.3	74.0 ± 7.6	<0.0001
Male sex (*n*,%)	273.0 (80.0)	38.0 (70.0)	0.47
Height, cm	161.7 ± 9.4	157.9 ± 8.9	0.006
Body mass index	23.8 ± 3.6	22.8 ± 2.8	0.03
Weight, kg	62.8 ± 13.2	57.2 ± 9.4	0.0002
Systolic BP, mmHg	131.2 ± 26.6	131.5 ± 25.2	0.95
Diastolic BP, mmHg	78.2 ± 18.3	71.5 ± 15.5	0.01
Heart rate, bpm	80.5 ± 21.9	83.6 ± 20.0	0.34
Blood test			
Red blood cell count, ×10^4^/mm^3^	7.1 ± 35.8	3.9 ± 0.8	0.1
Hemoglobin, g/dL	13.6 ± 2.0	12.2 ± 2.2	<0.0001
Hematocrit, %	40.0 ± 5.5	36.0 ± 6.2	<0.0001
Platelet count, ×10^4^/mm^3^	218.7 ± 68.7	227.9 ± 110.3	0.56
White blood cell count, /mm^3^	10.6 ± 4.3	9.2 ± 4.3	0.04
Lymphocytes, /mm^3^	2126.7 ± 1470.1	1708.0 ± 1248.1	0.05
Total bilirubin, mg/dL	0.7 ± 0.3	0.8 ± 0.5	0.22
AST, U/L	133.0 (25–72.3)	62.8 (25.5–54)	0.21
ALT, U/L	56.3 (17–42)	31.4 (15–29.5)	0.05
LDH, U/L	355.9 (199.8–347.5)	297.7 (202–335.5)	0.17
ALP, U/L	234.3 ±79.9	316.8 ± 250.7	0.03
γ-GTP, U/L	41.7 (18.3–47.8)	45.8 (17–46)	0.66
C-reactive protein, mg/dL	0.8 ±2.0	1.9 ± 4.9	0.1
Creatinine kinase, U/L	492.6 (104.8–421.3)	319.6 (81.5–356)	0.03
Creatinine kinase -MB, U/L	45.1 (6–44)	31.9 (7–33.5)	0.1
NT-pro BNP, pg/mL	402.1 (75.3–2013.7)	5993.1 (212–3720)	0.54
Blood urea nitrogen, mg/dL	18.5 ± 9.5	21.7 ± 15.8	0.16
Creatinine, mg/dL	1.1 (0.64–1.01)	1.6 (0.69–1.11)	0.14
eGFR, mL/min/1.73 m^2^	71.0 (54.3–88.5)	59.2 (46.8–74.7)	0.005
Sodium, mEq/L	139.3 ± 3.2	137.8 ± 4.0	0.01
Potassium, mEq/L	4.0 ± 0.6	4.2 ± 0.6	0.11
Chloride, mEq/L	104.0 ± 3.5	102.9 ± 3.8	0.03
Uric acid, mg/dL	5.9 ± 1.8	5.7 ± 1.8	0.37
Fasting plasma glucose, mg/dL	176.8 ± 79.1	164.6 ± 53.9	0.29
Hemoglobin A_1C_, %	6.3 ± 1.2	6.4 ± 0.9	0.76
Total protein, g/dL	6.6 ± 0.6	6.4 ± 0.9	0.12
Albumin, g/dL	3.7 ± 0.5	3.5 ± 0.6	0.005
Total cholesterol, mg/dL	187.1 ± 44.5	169.5 ± 58.7	0.04
Triglyceride, mg/dL	131.8 (67.3–160.8)	123.0 (65.3–146.8)	0.62
HDL-cholesterol, mg/dL	48.2 ± 11.7	47.6 ± 13.6	0.72
LDL-cholesterol, mg/dL	121.2 ± 37.6	104.7 ± 45.8	0.02
PT-INR	1.1 ± 0.4	1.1 ± 0.3	0.92
APTT, sec	55.7 (26.2–50.2)	51.8 (27.7–39.2)	0.63
D-dimer, μg/mL	3.9 (0.6–1.8)	4.8 (0.9–2.1)	0.69
FDP, μg/mL	10.2 (2.5–5.6)	12.1 (2.9–7.6)	0.66
Cardiac ultrasonography			
AOD, mm	30.1 ± 4.2	29.1 ± 4.3	0.18
LAD, mm	33.7 ± 6.1	33.5 ± 6.0	0.87
IVST, mm	9.8 ± 2.1	9.5 ± 1.9	0.39
PWT, mm	10.3 ± 5.9	9.9 ± 1.7	0.32
LVDd, mm	45.1 ± 6.9	44.2 ± 5.8	0.39
LVDs, mm	32.9 ± 7.2	32.2 ± 7.5	0.54
EF, %	50.6 ± 14.2	52.6 ± 14.2	0.33
Nutrition status			
CONUT score	1.9 ± 2.0	3.2 ± 2.6	0.002
mGPS			
score 0, *n* (%)	233 (57.3)	21 (5.2)	0.0002
score 1, *n* (%)	83 (20.4)	18 (4.4)	
score 2, *n* (%)	38 (9.3)	13 (3.2)	
GPS			
score 0, *n* (%)	249 (61.3)	28 (6.8)	0.003
score 1, *n* (%)	75 (18.5)	13 (3.2)	
score 2, *n* (%)	30 (7.3)	11 (2.7)	
Intravascular ultrasound			
CSA, mm^2^	14.7 ± 4.6	12.1 ± 2.9	<0.0001
MLA, mm^2^	4.1 ± 1.3	3.5 ± 1.3	0.04
PA (CSA-MLA), mm^2^	10.6 ± 4.2	8.6 ± 2.8	0.0001
Percentages of PA, % (%PA = (CSA-MLA)/CSA)	70.9 ± 9.7	69.7 ± 9.6	0.47

Data are mean ± SD or median (interquartile rage).Abbreviations: BP: blood pressure, AST: aspartate aminotransferase, ALT: alanine aminotransferase, LDH: Lactate dehydrogenase, γGTP:γ-Glutamyl transpeptidase, ALP: alkaline phosphatase, NT-pro BNP: N-terminal pro-brain natriuretic peptide, eGFR: estimated glomerular filtration rate, HDL: high-density lipoprotein, LDL: low-density lipoprotein, PT-INR: Prothrombin time-international normalized ratio, APTT: activated partial thromboplastin time, FDP: fibrin/fibrinogen degradation products, AOD: aortic dimension, LAD: left atrial dimension, IVST: interventricular septum thickness, PWT: posterior wall thickness, LVDd: left ventricular diameter at end diastole LVDs: left ventricular diameter at end systole, EF: ejection fraction, CONUT: Controlling Nutrition Status, mGPS: modified Glasgow Prognostic Score, GPS: Glasgow Prognostic Score, CSA: average reference lumen cross-sectional area, MLA: minimal lumen area, PA: plaque area.

**Table 2 nutrients-13-02663-t002:** Comorbidities and medicine of acute myocardial infarction patients with and without cancer.

	Non-Cancer (*N* = 358)		Cancer (*N* = 53)		*p* Value
	Yes, *n*	%	Yes, *n*	%	
Acute myocardial infarction	319	89.1	45	84.9	0.37
Unstable angina pectoris	39	10.9	8	15.1	0.37
**Responsible lesion**					
Left anterior descending artery	190	53.1	30	56.6	0.63
Left circumflex artery	36	10.1	8	15.1	0.27
Multivessel disease	174	48.6	32	60.3	0.23
Right coronary artery	125	34.9	15	28.3	0.34
Smoking (Current and former)	229	64.0	35	66.0	0.76
**Comorbidities**					
Hypertension	265	74.0	41	77.4	0.60
Dyslipidemia	263	73.5	33	62.3	0.09
Diabetes mellitus	155	43.3	32	60.4	0.02
Hyperuricemia	91	25.4	13	24.5	0.89
Chronic kidney disease	117	32.7	25	47.2	0.04
Hemodialysis	13	3.6	4	7.5	0.18
Percutaneous coronary intervention	48	13.4	9	17.0	0.48
Coronary artery bypass graft	5	1.4	3	5.7	0.03
Aortic disease	10	2.8	3	5.7	0.26
Collagen disease	11	3.1	4	7.5	0.11
Peripheral artery disease	16	4.5	6	11.3	0.04
Cerebrovascular disease	45	12.6	10	18.9	0.21
**Medication**					
Angiotensin-converting enzyme inhibitor	25	7.0	5	9.4	0.52
Angiotensin II receptor blocker	102	28.5	15	28.3	0.97
Aspirin	65	18.2	15	28.3	0.08
Beta blocker	43	12.0	9	17.0	0.31
Antihyperuricemic	19	5.3	4	7.5	0.51
Calcium channel blocker	116	32.4	19	35.8	0.62
Diuretic	37	10.3	8	15.1	0.30
Clopidogrel	34	9.5	10	18.9	0.04
Prasugrel	3	0.8	0	0.0	0.50
Ticlopidine	1	0.3	2	3.8	0.01
Warfarin	9	2.5	3	5.7	0.20
Direct oral anticoagulants	5	1.4	0	0.0	0.39
Other antiplatelet agents	10	2.8	5	9.4	0.02
Dipeptidyl-peptidase IV inhibitor	54	15.1	13	24.5	0.08
Sodium glucose cotransporter II inhibitor	6	1.7	1	1.9	0.91
Insulin	20	5.6	4	7.5	0.57
Other oral hypoglycemic agent	59	16.5	12	22.6	0.26
Statin	83	23.2	16	30.2	0.27
Omega-3 fatty acid ethyl esters	4	1.1	1	1.9	0.63
Eicosapentaenoic acid	12	3.4	3	5.7	0.40
Ezetimibe	7	2.0	2	3.8	0.40
Anti-cancer agent	0	0.0	8	15.1	<0.0001
Immunosuppressant	6	1.7	2	3.8	0.30
Steroid	11	3.1	5	9.4	0.03
In-hospital mortality	17	4.7	4	7.5	0.38
Intravascular ultrasound	304	84.9	47	88.7	0.47

**Table 3 nutrients-13-02663-t003:** Divided groups by age and atherosclerotic plaque area.

Group	N	Cancer	Age	Plaque Area	Group Definition
G1	90	No	<60 years old	N/A	Non-cancer/Low risk
G2	122	No	≥60 years old	≥9.39 mm^2^	Non-cancer/Middle risk
G3	146	No	≥60 years old	<9.39 mm^2^	Non-cancer/High risk
G4	53	Yes	≥60 years old	N/A	Cancer

**Table 4 nutrients-13-02663-t004:** Comportments of Synthetic variable with weights.

Synthetic Variable	Original Variables with Its Weight			
**Nutrition**	CONUT (0.243)	GPS (0.898)	mGPS (0.846)	cons * (−1.307)
**Lipid**	T.chol (0.015)	LDL-c (0.018)	-	cons * (−4.947)
**Glucose**	FPG (0.009)	HbA1c (0.594)	-	cons * (−4.45)
**Blood pressure**	sBP (0.027)	Pulse pressure (0.038)	-	cons * (−5.559)
**Renal function**	Cr (0.469)	eGFR (0.025)	-	cons * (1.151)

Abbreviations: CONUT; Controlling Nutrition Status, GPS; Glasgow Prognostic Score, mGPS; modified Glasgow Prognostic Score, T.chol; total cholesterol, LDL-c; low-density lipoprotein cholesterol, FPG; Fasting plasma glucose, HbA1c; hemoglobin A1c, sBP; systolic blood pressure, Cr; Creatinine, eGFR; estimated glomerular filtration rate, cons *; constant term used to calculate principal component score along with weight of each variable.

**Table 5 nutrients-13-02663-t005:** Mean and SD of six risk variables by Four Groups.

	G1	G2	G3	G4	*p* Value *
Nutrition	−0.65 (1.13)	−0.15(1.43)	0.26(1.68)	0.76(1.85)	<0.0001
Lipid	0.44 (1.33)	0.001(1.19)	−0.09(1.39)	−0.50(1.69)	0.001
Glucose	−0.06 (1.15)	−0.09(0.95)	0.15(1.30)	−0.11(0.81)	0.27
Uric acid	6.33 (1.55)	5.86(1.76)	5.75(1.91)	5.70(1.82)	0.13
Blood pressure	−0.14 (1.23)	−0.01(1.38)	0.01(1.24)	0.22(1.49)	0.47
Renal function	−0.60 (0.83)	0.01(1.07)	0.19(1.36)	0.47(1.72)	<0.0001

*p* value * based on One-Way Analysis of Variance (ANOVA). Data are means (SD). The equation of nutrition was expressed as “Nutrition = (−1.307) + 0.243 * CONUT + 0.898 * GPS + 0.846 * mGPS”. Similarly, other equations were expressed as “Lipid = (−4.947) + 0.018 * LDL-c + 0.015 * T.chol”, “Glucose = (−5.448) + 0.594 * HbA1c + 0.0092 * FPG”, “Blood pressure = (−5.559) + 0.038 * Pulse pressure + 0.027 * systolic BP”, “Renal function = (1.151) + 0.469 * Cr + (−0.025) * eGFR”. Nutrition, lipid, and renal functions were significant predictors of risk grouping.

**Table 6 nutrients-13-02663-t006:** Estimate of the ordinal logistic regression model.

Parameter	Estimate	SE	Wald χ2	*p* Value
G4 (α1)	−0.985	0.362	7.39	0.01
G3 (α2)	1.157	0.359	10.39	0.00
G2 (α3)	2.658	0.376	50.00	<0.0001
Nutrition	0.232	0.071	10.64	0.00
Lipid	−0.134	0.080	2.80	0.09
Glucose	0.032	0.087	0.14	0.71
Blood pressure	0.157	0.074	4.53	0.03
Renal	0.363	0.091	16.00	<0.0001
Uric acid	−0.209	0.059	12.59	0.00
smoking (Yes)	−0.081	0.100	0.65	0.42

**Table 7 nutrients-13-02663-t007:** Estimates of odds ratio from ordinal logistic regression model.

Variable	Odds Ratio *	95% CI	*p* Value
Nutrition	1.26	1.10–1.45	0.001
Lipid	0.88	0.75–1.02	0.09
Glucose	1.03	0.87–1.23	0.71
Uric acid	0.81	0.72–091	0.0004
Blood pressure	1.17	1.01–1.35	0.03
Renal function	1.44	1.20–1.72	<0.0001
Smoking (Yes vs. No)	0.85	0.58–1.26	0.42

Odds ratio *: G1 is used as reference group.

**Table 8 nutrients-13-02663-t008:** Estimates of odds ratio from ordinal logistic regression model.

Variable	Odds Ratio *	95% CI	*p* Value
Nutrition	0.927	0.709–1.211	0.578
Lipid	0.700	0.477–1.026	0.068
Glucose	0.860	0.573–1.291	0.467
Blood pressure	1.002	0.753–1.334	0.987
Renal function	1.316	0.976–1.775	0.072
Uric acid	0.972	0.782–1.209	0.800
Smoking (Yes vs. No)	1.419	0.626–3.217	0.403
Age	1.036	0.996–1.079	0.079
Plaque area	0.855	0.759–0.964	0.011

## Data Availability

The data presented in this paper are available on request from the corresponding author.

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
