# Peer review of "Nutrition Status and Renal Function as Predictors in Acute Myocardial Infarction with and without Cancer: A Single Center Retrospective Study"

_nutrients, 2021, doi:10.3390/nu13082663_

Round 1

Reviewer 1 Report

This manuscript investigated differences in clinical parameters in acute myocardial infarction (AMI) with and without the history of cancer and found that nutrition status and renal dysfunction were associated with AMI with cancer. The study also presented an equation to predict the presence of cancer in AMI patients, which is less convincing. Clarity of presentation also needs improvement.

Specific comments:

  1. The study divided patients into synthetic risk groups and age was an important factor in determining the synthetic risk. This may not represent the true risk. Then, ordinal logistic regression was used to evaluate the odds ratios of nutrition, lipid, glucose, blood pressure, and renal function, but these metrics are highly likely to be age-dependent. As a result, the reported odds ratios were partially related to age, which is less convincing. It is suggested to only predict cancer vs non-cancer groups.
  2. The authors employed a logistic regression model to predict the probability of cancer. This seems to represent only the descriptive/fitting capability of the model since the entire dataset was used in the logistic regression, but the predictive capability was not evaluated by using a separate “test” dataset. At least, the term “predictive equation” needs to be reworded.
  3. Section 2.7, please list the equations separately from text and number the equations.
  4. Figure 4: only means were plotted. Is it possible to add standard deviations?

Author Response

Responses to the Reviewer 1

Reviewer 2 Report

In this manuscript, Itaya, et. al., have reported a retrospective analysis of characteristics of patients with acute myocardial infarction (AMI) to develop a predictive metric for incidence of cancer. To develop this predictive model, they have utilized data from more than 400 patients and performed ordinal regression analysis to model various co-morbidities. They show that nutrition status (as evaluated by three different metrics – CONUT, GPS and modified GPS) and renal function are closely associated with AMI/cancer and that nutrition status, in particular, is the primary differentiator between patients with and without cancer.

Some comments:

(1) There are very few female patients in this cohort (80% male in non-cancer and 70% male in cancer groups). Would it not be a good idea to perform this analyses only on male patients? With such skewed sex ratios, it would hard to draw any gender differences and it may just be better to leave out female patients in this study. Alternatively, they could attempt to stratify the results by sex and see if primary conclusions are valid for both male and female patients – with the caveat that the sample size is small for female group. If such an analysis cannot be carried out, please include this aspect under limitations.

(2) Line 124 (section 2.7) mentions six synthetic variables but Table 4 (page 8) shows only five synthetic variables. Tables 5 and 6, however, show six synthetic variables again. Please address this.

(3) Along the lines of the previous comment – Line 123 defines vector of covariates as X = (X1, X2, … Xn=6, W) where W is 0 or 1 for smoking habits. This is completely ignored in rest of the manuscript. According to Table 1, ~65% of patients were smokers. How did the results compare between smokers and non-smokers?

(4) Line 127: Why is α4 = 0? Please show the parameter vector β explicitly?

(5) It is unclear how the odds ratios were calculated. It is mentioned under Table 5 that “G1” was used as reference group. Please explain calculation of odds ratio in much greater detail.

(6) More details on classification and regression tree analysis would be useful. Isn’t 9.39 mm2 plaque quite large? Also, did the plaques fall under the same classification?

Minor issues:

(1) In section 2.7, please do not use inline equations – particularly the predicted probability equations. Please list equations as their own lines and number them.

(2) Lines 199 – 205: please list equations sequentially in separate lines and number them.

(3) Lines 246 – 248: It is not clear what the authors are referring to in these lines. Are the authors attributing the higher incidence of cancer in this cohort to hospitalization at later stages of diseases?

(4) It would nice to add another paragraph in the introduction to better explain the focus of the study. It was not clear that the authors are developing a predictive metric for incidence of cancer from the introduction.

(5) There are several typographical errors that need to be corrected.

Author Response

Responses to the Reviewer 2

Round 2

Reviewer 1 Report

The manuscript was significantly improved.

Reviewer 2 Report

Authors have addressed my comments thoroughly.